# The Role of E3, E4 Ubiquitin Ligase (UBE4B) in Human Pathologies

**DOI:** 10.3390/cancers12010062

**Published:** 2019-12-24

**Authors:** Nikolaos Antoniou, Nefeli Lagopati, Dimitrios Ilias Balourdas, Michail Nikolaou, Alexandros Papalampros, Panagiotis V. S. Vasileiou, Vassilios Myrianthopoulos, Athanassios Kotsinas, Yosef Shiloh, Michalis Liontos, Vassilis G. Gorgoulis

**Affiliations:** 1Molecular Carcinogenesis Group, Department of Histology and Embryology, School of Medicine, National Kapodistrian University of Athens, 75 Mikras Asias Str., Goudi, GR-11527 Athens, Greece; nik_biol@hotmail.com (N.A.); nlagopati@med.uoa.gr (N.L.); panagiotis.vasileiou@yahoo.gr (P.V.S.V.); mliontos@gmail.com (M.L.); 2Department of Pharmacy, National Kapodistrian University of Athens, Panepistimiopolis Zografou, GR-15771 Athens, Greece; jimdas1@gmail.com (D.I.B.); vmyriant@pharm.uoa.gr (V.M.); 3General Maternal Hospital of Athens “Elena Venizelou”, GR-11521 Athens, Greece; nikolaoumike@hotmail.com; 4First Department of Surgery, Laikon Teaching Hospital, School of Medicine, National Kapodistrian University of Athens, 75 Mikras Asias Str., Goudi, GR-11527 Athens, Greece; a_papalampros@hotmail.com; 5The David and Inez Myers Laboratory for Cancer Research, Department of Human Molecular Genetics and Biochemistry, Sackler School of Medicine, George S. Wise Faculty of Life Sciences, Tel Aviv University, Tel Aviv 69978, Israel; yossih@tauex.tau.ac.il; 6Oncology Unit, Department of Clinical Therapeutics, Medical School, National and Kapodistrian University of Athens, Alexandra Hospital, GR-11528 Athens, Greece; 7Biomedical Research Foundation of the Academy of Athens, GR-11527 Athens, Greece; 8Faculty of Biology, Medicine and Health, University of Manchester, Manchester Academic Health Science Centre, Manchester M20 4GJ, UK

**Keywords:** UBE4B, ubiquitin ligases, DNA damage response, cancer disease, p53

## Abstract

The genome is exposed daily to many deleterious factors. Ubiquitination is a mechanism that regulates several crucial cellular functions, allowing cells to react upon various stimuli in order to preserve their homeostasis. Ubiquitin ligases act as specific regulators and actively participate among others in the DNA damage response (DDR) network. UBE4B is a newly identified member of E3 ubiquitin ligases that appears to be overexpressed in several human neoplasms. The aim of this review is to provide insights into the role of UBE4B ubiquitin ligase in DDR and its association with p53 expression, shedding light particularly on the molecular mechanisms of carcinogenesis.

## 1. Introduction

Ubiquitination is a mechanism that regulates several cellular functions, such as growth, DNA repair, transcriptional regulation, intracellular signaling, autophagy, cell cycle, and programmed cell death [1,2,3,4,5,6]. Consequently, ubiquitination allows cells to react upon various stimuli aiming to preserve their homeostasis [7].

The mammalian ubiquitin (Ub) chain assembly factor, or ubiquitination factor E4B (UBE4B/UFD2a), belongs to E3 ubiquitin ligases and is located in the chromosome 1p [8,9,10]. Accumulating evidence implicates UBE4B in DNA double-strand break (DSB) repair and as a regulator of p53, a key effector of the DNA damage response and repair (DDR/R) pathway. In the current manuscript, we provide an overview on the roles of UBE4B/UFD2a in normal cellular function and its implication in human disorders such as cancer development.

## 2. Ubiquitination and Ubiquitin Proteasome System

Protein homeostasis is fundamental in normal cellular function and cell survival [11]. Ubiquitin Proteasome System (UPS) is the major pathway for the physiological turnover of short-lived intracellular proteins [12,13]. Thus, the UPS plays an important role in maintaining the protein homeostasis network via selective elimination of damaged or misfolded proteins. Impaired function of UPS is implicated in the physiological aging process and also in several age-related disorders, such as neurodegenerative diseases and cancer, which are characterized by increased accumulation of oxidized proteins and protein aggregates [11,14].

A key factor in UPS is ubiquitin, a small but essential protein of 76 residues (7.7 kDa), with a five-stranded antiparallel beta-sheet traversed by a single helix, and it owes its name to the fact that it is ubiquitously found in all eukaryotic cells. The C-terminal carboxylate group of G76 of ubiquitin seems to play a crucial role in the multistep process of ubiquitination [5].

Degradation of proteins through UPS requires two major successive steps, the covalent attachment of multiple molecules of ubiquitin to the target protein and the degradation of ubiquitinated proteins by the 26S proteasome [15,16,17]. Lysine residues on the target protein are closely associated with conjugation of the ubiquitin moieties through a covalent link [18]. The formation of a poly-ubiquitin chain includes the addition of at least four ubiquitin molecules that sequentially conjugate on the K48 residue of each ubiquitin molecule and are necessary for the target protein to be recognized by the 26S proteasome [5,19,20].

Three classes of enzymes are consecutively required for the polymerization of ubiquitin as a signaling label on a substrate, the E1s, E2s and E3s [19,21,22]. The first step of ubiquitination involves the activation of ubiquitin by the formation of a covalent bond with ubiquitin-activating enzymes, E1. During the second step, E1s deliver the activated ubiquitin molecule to the E2 ubiquitin-conjugating enzymes, where they are conjugated by a thioester bond. Finally, E3 ubiquitin-protein ligases catalyze the transfer of ubiquitin from E2s to a lysine residue in the protein substrate [23,24,25]. Ubiquitin E3 ligases can be classified in three main types depending on the presence of characteristic domains and on the mechanism of ubiquitin transfer to the protein substrate. These comprise the HECT E3s (Homologous to E6-AP Carboxyl Terminus), RING E3s (Really Interesting New Gene) and the RBR E3s (RING-between RING-RING) [26,27].

E3 ligases have been found to act by either promoting cancer progression or suppressing it, depending on the cellular environment [28]. The E3 ubiquitin ligase BRCA1 is a well-established tumor-suppressor protein usually found mutated in breast cancer [29]. Another well studied E3 ligase is RNF168, which is thought to orchestrate DSBs repair, favoring the Non-Homologous End Joining (NHEJ) mechanism through the regulation of 53BP1 [30]. Overexpression of this ligase in various cancer cell lines has been shown to be essential for the retention of 53BP1 foci on chromatin upon a proteotoxic crisis due to proteasome inhibition. The latter phenomenon results in redirecting DSBs repair from the Homologous Recombination pathway to NHEJ, emergence of genomic instability and tumor progression [31].

An additional well-studied category of ubiquitin ligases is the class of E4s, which play multiple roles. Their action is either complementary to E3s, particularly the RING E3s, by enhancing the length of polyubiquitin chains, or under certain circumstances simulating the function of E3 ligases [32,33]. A specific feature of RING E3s is the presence of a zinc-binding domain or a U-box domain. Both E3s and E4s containing U-box domains can function either as monomers or homodimers [26,27].

The mammalian Ub chain assembly factor, or ubiquitination factor E4B (UBE4B/UFD2a), belongs to E3 ubiquitin ligases, contains a conserved U-box catalytic domain of about 70 amino acids, and functions as a monomeric protein [8,9,10]. The aforementioned domain retains the same fold as the RING catalytic domain without bearing the zinc groups and mediates the interaction of this factor with the ubiquitin-charged E2 enzyme to promote the attachment of a poly-ubiquitin chain on a selected target of UBE4B, based on its E3 and E4 ligase activity [8,9,10,26]. Actually, UBE4B and its isoform UBE4A belong to a new class of ubiquitination enzymes representing the U-box-containing RING family of ubiquitin ligases [34]. UBE4A is able to facilitate the recruitment of proteins related to the homologous recombination repair pathway [35]. Both UBE4A and UBE4B are required in the degradation of certain types of substrates through a Ub fusion degradation pathway, designated as UFD, in the same way as the yeast orthologue UFD2 [36,37,38]. Furthermore, both the mammalian UFD2a and the yeast UFD2 proteins bind to the Ub moieties of preformed conjugates, catalyzing the elongation of the polyubiquitin chain [36,37,38]. The yeast UFD1, UFD2, UFD3, UFD4, and UFD5 homologues are among the most important genes that are involved in the UFD pathway [36]. Ub-mediated proteolysis is considered essential for normal growth. It is implicated in various biological processes, such as cell differentiation, response in stress stimuli, cell cycle control, regulation of transcription, and programmed cell death [1,2,3,4,5,6].

UBE4B ligase coordinates DNA double-strand break repair and apoptosis induction in *Caenorhabditis elegans* [39]. UBE4B forms focal accumulations after DSB’s induction upon Ionizing Radiation (IR) which is independent of its ligase activity. Thus, UBE4B seems to play a key role as part of a *bona fide* cellular network which facilitates the communication between DNA repair and apoptotic response [39]. Notably, unpublished data from our team showed the nuclear presence of UBE4B foci also in HCT116 (human colorectal carcinoma cell line) and co-localization with DSBs following Doxorubicin administration (Figure 1).

According to recent studies, UBE4B physically interacts with wild type p53 [40]. The stability of p53 is determined by the RING domain E3 ubiquitin ligases Mdm2, Pirh2 and COP1 [41,42]. Particularly, Mouse Double Minute 2 (MDM2) is often overexpressed in various types of cancer, inducing proteasome-mediated degradation of p53, thus endorsing cell survival and proliferation [43,44,45]. UBE4B also interacts with MDM2 and is essential for MDM2-mediated p53 poly-ubiquitination and degradation. Therefore, an increased UBE4B activity may be considered as an oncogenic feature, by inhibiting the activity of p53 in cancer cells and promoting tumorigenesis [46]. The increased UBE4B levels in various tumors supports this notion as described in the following section. Notably, studies have demonstrated that the regulatory role of UBE4B is not restricted only to p53 but includes also its family members, namely p63 and p73, as further described in the next section [9,47].

## 3. UBE4B and p53 Family Members

The stability of p53 is primarily controlled by MDM2, which targets p53, leading to proteasomal degradation. The U-box catalytic domain of UBE4B is closely related to the RING-finger domain of MDM2 and is responsible for its E3 activity. These two enzymes are considered to be significant regulators of p53 through the ubiquitination process [48,49,50]. Loss of MDM2 leads to an activation of p53 that is lethal during embryogenesis. The pathways that allow stress-induced inhibition of MDM2 are essential to activate the cell growth suppressive activity of p53 [51]. Failures in the pathways that control MDM2 switch-off have been linked to cancer development [52,53].

In certain cases, MDM2 does not have the ability to functionally silence p53 on its own. The latter property is usually associated with higher levels of Mdm2 and takes place inside the nucleus [54,55]. In this aspect, there is important evidence that UBE4B can promote the poly-ubiquitination of p53 in a synergistic manner with MDM2, inhibiting cell apoptosis and promoting tumorigenesis [38,46,54,56]. Actually, UBE4B interacts directly with MDM2 and this interaction reduces the half-life of p53 via proteasome-mediated degradation, leading to repression of p53-dependent transactivation and apoptosis [46]. Moreover, in the presence of DSBs, UBE4B is able to negatively regulate the protein levels of two phosphorylated active forms of p53 namely phospho-p53 (Ser15) and phospho-p53 (Ser392) independently of MDM2. This action is mainly performed within the nucleus [57].

Apart from its well-studied tumor-suppressor activity, p53 is also an important factor in the development of the nervous system [58]. P53 overexpression has been reported in many neurodegenerative conditions, including Parkinson’s, Huntington’s and Alzheimer’s diseases, seizure-induced excitotoxic damage, middle cerebral artery occlusion, traumatic brain injury, and peripheral nerve injury [47,59]. Through its regulatory action on the p53 family proteins, UBE4B can act as a key factor in the development of the nervous system and might possibly comprise a relevant druggable target (see below) in various neurodegenerative diseases [47]. Interestingly, UBE4B has been found to poly-ubiquitinate an abnormal form of ataxin-3 which is responsible for the development of Machado-Joseph disease, thereby marking it for degradation by the ubiquitin-proteasome pathway [60]. Ataxin-3 is implicated in DNA damage repair and plays a crucial role in neuronal development [61].

The protein p63 has been identified as a homolog of the tumor suppressor protein p53 and it has an important implication in tumor development by regulating apoptosis, while it is also capable of inducing cell cycle arrest and cellular senescence [62]. Both p63 and p53 target similar pro-apoptotic proteins. This is exemplified among others by their common ability to increase the expression of Bax, an essential factor for activation of the mitochondrial apoptotic pathway [63]. Moreover, the presence of p63 has been suggested as an indispensable requirement for the pro-apoptotic activity of p53 [64]. This has been strengthened by previous findings that cells deficient for both p63 and p73, exhibit significant resistance to neuronal apoptosis despite the presence of functional p53 [47]. Is UBE4B specifically regulating the ubiquitination on p63, promoting p63′s proteasomal-mediated degradation? In this case, however, the mechanism is more complex. Only one isoform has been found to be regulated by UBE4B, namely ΔNp63α, whereas the TA isoform was not linked to UBE4B [65]. ΔNp63α, the dominant negative isoform of the p63 family, is an essential survival factor. UBE4B binds to ΔNp63α isoform and stabilizes it. This stabilization is achieved via the inhibition of ubiquitination of the latter [65]. The fact that UBE4B has no relation to any of the β isoforms of either p63 or p73 might also explain the specificity of this ligase [47].

Another human p53-related protein is p73 [66]. It is usually activated after DNA damage in a way that is distinct from that of p53 [67]. When overproduced, it can activate the transcription of p53-responsive genes and inhibit cell growth in a p53-like manner by inducing apoptosis [68,69]. In particular, p73 transactivates a large number of p53 target genes such as p21 and Bax [67,69]. Interestingly, it has been shown that p53 is able to induce cell cycle arrest in the absence of p73 but not apoptosis [70]. Unlike p53, p73 encodes for different isoforms [71,72]. UBE4B has been shown to bind to p73α but not to p73β isoform. UBE4B can also induce p73α proteasomal degradation, independently of ubiquitin-conjugation [9]. Additionally, p73α binds to MDM2 without being targeted for degradation. Through this interaction, the function of p73 as an apoptotic factor is suppressed due to its sequestration, and thus, its inability to cooperate with its transcriptional co-factor p300/CBP [73].

In summary, p53 turnover is regulated by both MDM2 and UBE4B in a cancer cell. Mono-ubiquitination of p53 on single or multiple sites is usually followed by a poly-ubiquitination of these sites through UBE4B. The elongated ubiquitin chain renders the modified p53 molecule recognizable by the proteasome. This process reduces the protein expression levels and transcriptional activity of p53 in the cytoplasm and in the nucleus, respectively (Figure 2a,b). Based on the proposed model of oncogene-induced cancer development [74], p53 is a key effector of the DDR and therefore UBE4B can be considered as an oncogene due to the down-regulation of p53. Moreover, some reports indicate the contribution of UBE4B towards cancer cell survival upon DNA damage induced by factors such as chemical agents or IR. This is also linked with the well-studied interaction between UBE4B and p53 family members [9,57,65]. A recent study not only supports this notion but also focuses on the role of mir-1301 as a novel negative regulator of UBE4B. Actually, this micro-RNA potentially halts cancer cell migration and metastasis via stabilization of p53 [75]. Despite the growing amount of evidence which implicate UBE4B in cancer development, the precise regulation of UBE4B over p53 and how this could drive cancer progression requires further investigation. Moreover, according to recent findings, UBE4B interacts physically with p73a isoform in various cancer cell lines, hindering the tumor-suppressor activity of p73a by promoting its proteasomal degradation (Figure 2c). This potential tumorigenic role of UBE4B has been confirmed in head, neck and lung cancer cell lines. In addition, it seems that UBE4B weakens ubiquitination of ΔΝp63a (Figure 2d). On the other hand, a potential interaction of UBE4B with other p53 family members such as p73β, ΔΝp73 and p63β isoforms has not yet been confirmed and whether these isoforms are indeed influenced by this ligase remains to be elucidated (Figure 2e).

## 4. UBE4B Ubiquitin Ligase in Human Pathology

The ubiquitin ligases can maintain cellular viability and homeostasis [76,77]. The deletion of UBE4B in vivo is likely to lead to early embryonic death, due to the induction of apoptosis in the heart, an organ where it is exclusively expressed during this developmental stage [55,78]. Intriguingly, UBE4B is involved in a limited number of pathological conditions, which mostly represent neuropathies and cancer, whereas alterations either in the *UBE4B* gene or in the protein have been frequently found in various types of cancer (see Table 1).

### 4.1. UBE4B and Neurodegenerative Diseases

Normally, UBE4B is a factor regulating the development of the nervous system and can be a target molecule in neurodegenerative disease treatments [47]. UBE4B has been found to be associated with the poly-ubiquitination of an abnormal form of ataxin-3, which has been shown to be responsible for the development of Machado-Joseph disease (MJD) [60]. MJD is one of approximately 30 recognized, dominantly inherited forms of ataxia [61], which is generally characterized by a lack of muscle control or coordination. The neuroprotective role of UBE4B has also been shown by its mouse orthologue Ube4b/Ufd2a in Wallerian degeneration process [47]. The latter occurs upon an injurious stimulus and leads to axonal death [83]. However, a chimeric mutation between the genetic locus of *Ube4b* and nicotinamide mononucleotide adenylyl transferase 1 (*Nmnat1*), observed in mice, has been found to delay the Wallerian degeneration [83]. In addition, in the inclusion body myopathy with Paget disease of bone and frontotemporal dementia (IBMPFD), p97 chaperon is mutated and the interaction of this mutant protein with the UBE4B ligase is defective compared to its interaction with ataxin-3 which is enhanced. Consequently, the abnormal aggregates composed of p97 and ataxin-3 account for the neurodegenerative manifestations of this proteinopathy [80]. Overall, it appears that UBE4B has a beneficial impact on the function of the nervous system. Intriguingly, it is rarely affected in neurodegenerative diseases, suggesting that other interacting molecular partners dictate its malfunction in these pathological conditions.

### 4.2. UBE4B and Cancer

UBE4B has been found either overexpressed or suppressed in diverse types of human cancer [10,38,46,56,81,82]. It is noteworthy that a 500-kb genetic locus in the small arm of chromosome 1 is frequently mutated in neuroblastoma (NBL) cases [10,81]. Deletion of this region within 1p36.2-3 has been linked with the inactivation of tumor suppressors including the UBE4B ligase [10,81]. Moreover, a variety of single nucleotide polymorphisms (SNPs) have been described within the coding region of the *UBE4B* gene, but they do not seem to affect NBL development [10,81]. Among them, only the c.1439 + 1G > C SNP was found to affect a splice donor site mutation of exon 9 and was associated with a poor outcome of stage 3 patients [10,81]. Three potential regions within 1p36.1-3 susceptible to deletions have been linked with Oral Squamous Cell Carcinoma progression [82]. On the other hand, brain tumors such as medulloblastoma and ependymoma exhibit elevated protein levels of UBE4B which are often attributed to increased levels of mRNA, whereas sometimes augmentation of mRNA levels is related to gene amplification [46]. Interestingly, the observed loss of p53 function in brain tumors could be due to the negative regulation from UBE4B, leading p53 to an MDM2-mediated degradation through the proteasome [46]. UBE4B protein has also been reported to be overexpressed in breast cancer tissues, while both protein and mRNA levels are increased in hepatocellular carcinomas [38,56]. Preliminary unpublished data from our group also clearly indicate increased levels of this ligase in Non-Small Cell Lung Cancer (NSCLC) tissues compared to their normal counterparts (Figure 3a,c). This difference in expression is observed regardless of the genetic status of p53 (either wild type or mutated) (Figure 3a). However, UBE4B seems to be expressed at higher levels in a wt-p53 environment in contrast to cases with mutant p53 (Figure 3a,b). Upon monitoring of UBE4B expression in colon cancerous tissues, we found the same expression pattern detected in lung cancer, which is the upregulation of UBE4B ligase in malignant areas relative to the normal ones.

It is worth noting that the expression of UBE4B gradually increased during precancerous colorectal lesions, with its highest levels being present in high grade dysplasia (Figure 4). This is an interesting and novel finding because it constitutes an evidence for the implication of UBE4B in carcinogenesis from its early stages. This finding may be related to its role as a negative regulator of wtp53, but this merits further exploration.

Only a few studies have reported oncogenic behavior of UBE4B, which relies on its forced overexpression in cancer cell lines and animal models, leading to larger tumors compared to cancer cells bearing solely the endogenous gene. This is achieved through the interaction of the ligase with wtp53, decreasing the apoptotic and transcriptional activity of the latter [38,46,56]. Our above in-situ findings from human clinical samples support that UBE4B is a potentially significant regulator of cancer progression. The sporadic overexpression of UBE4B in pre-cancerous lesions of human colon tissues, even from the phase of hyperplasia, reinforces this hypothesis. Nevertheless, additional research in this field is needed in order to draw a safe conclusion.

Recently, certain studies suggested a possible relationship of UBE4B with DNA damage occurrence. Particularly, it has been shown that gamma irradiation, a known DSB inducer, leads to increased UBE4B protein levels in various human cancer cell lines. This is followed by a boosted poly-ubiquitination of wtp53 that is necessary for its proteasomal degradation [57].

Notably, Du et al. observed that the levels of UBE4B in the nucleus are significantly decreased in response to DNA damage upon IR, while MDM2 expression is increased. In this case, the affinity between UBE4B and MDM2 is greatly decreased following DNA damage [57]. Up to now, MDM2 was considered as the main E3 factor required for cooperating with UBE4B in order to degrade wtp53 under normal conditions [38,46,56]. However, it has been shown that this is not the case upon a stress stimulus (e.g., DSBs), since UBE4B has been found to be overexpressed at different time points relative to the MDM2 E3 ligase, possibly ubiquitinating wtp53 independently from MDM2 [57].

These findings suggest differential ways on how phosphorylated p53 may be regulated in response to DNA damage [57]. Our preliminary in-situ data on the UBE4B expression levels during early and late stages of carcinogenesis possibly may reflect this effect. Therefore, these results could also be considered as tenable indications, which would enable and encourage further investigation of the role of UBE4B in molecular carcinogenesis.

## 5. Druggability Analysis

The important array of functions of UBE4B highlights its significance as a potential drug target. To theoretically evaluate the protein druggability, three diverse technologies were utilized, namely the Sitemap mapping algorithm [84]; CavityPlus, a web server for binding site detection and druggability prediction [85]; and fpocket, a protein cavity detection web platform [86]. A homology model of human UBE4B was generated, implementing the structure of *Saccharomyces cerevisiae* ubiquitin ligase Ufd2p (similarity: 29.9%; positives: 49.7%; pdb entry: 2qiz) as a template. Human UBE4B (O95155) sequence was obtained from the Uniprot database [87]. The UniProt Consortium 2019) and Schrödinger Prime multiple sequence viewer tool were used to construct a structure-based alignment. Manual adjustments of the template [88] alignment were performed to remove helical gaps and to correct other helical alignment errors generated during the alignment procedure. The constructed homology model was used for further evaluation and druggability analysis simulation.

Druggability data were collected and compared. Interestingly, the above algorithms yielded overlapping results for certain protein sites, as shown in Figure 5, indicating high probability of druggable protein cavities existence. Specifically, the protein cavity depicted in Figure 5a appeared among the top ranks in all programs (Sitemap-SiteScore: 1.007; D-score: 1.045; threshold for druggable sites: 0.80; CavityPlus-DrugScore: 920.00; Druggable; fpocket-PocketScore: 10.04), while cavities illustrated in Figure 5b,d, scored high on Sitemap and CavityPlus (SiteScore: 1.007; D-score: 1.045; threshold for druggable sites: 0.80; DrugScore: 4689; Druggable) and Sitemap and fpocket (SiteScore: 1.010; D-score: 1.043; PocketScore: 10.13) respectively. It is also noteworthy that the top-scored cavity by the Sitemap algorithm (SiteScore: 1.293; D-score: 1.368), demonstrated in Figure 5c, is adjacent to the C757 residue, which corresponds to C385 in Ufd2p, whose C385Y mutation was found to disrupt in-vivo binding to Cdc48 [89] an ATPase with essential regulatory functions. Additional important residues of Ufd2p candidates for binding to Cdc48p are R844 and E855 [90], which correspond to R1191 and E1202 in UBE4B, respectively, and are colored black in Figure 5.

The homology model of human UBE4B provided adequate structural insight, thus affording opportunities for further structural exploitation of each druggable cavity in order to design potent, high-selectivity inhibitors. These sites can be used in virtual screening of large compound collections that may be used to advance the identification of molecules with potential UBE4B-inhibitory properties.

## 6. Conclusions and Future Perspectives

Ubiquitin ligases play a crucial role in the maintenance of cellular viability and homeostasis [76,77]. In some case they can act as oncogenes, promoting cancer progression when their degradation process is deregulated, while in other cases, ubiquitin ligases can behave as tumor suppressors.

UBE4B ligase activity has been demonstrated to be involved in cell cycle arrest during DDR [46,57,91,92,93]. Moreover, by regulating the extent of protein ubiquitination at DNA damaged sites, it contributes indirectly to the DDR. Nonetheless, the involvement of UBE4B in the DNA repair mechanisms has been shown mainly in lower eukaryotes like yeasts and worms, leaving an open window for further investigations [94,95,96].

Overall, although UBE4B is linked to different diseases and particular to cancer, the exact mechanism which is associated with the role of this molecule in the development of a cancer cell is still not clearly understood. Nevertheless, clinical data support an oncogenic activity based on the observed UBE4B overexpression in several neoplasms, such as in lung, colorectal, and head and neck squamous cell carcinoma [46,97], providing an opportunity to therapeutically target this molecule.

## Figures and Tables

**Figure 1 cancers-12-00062-f001:**
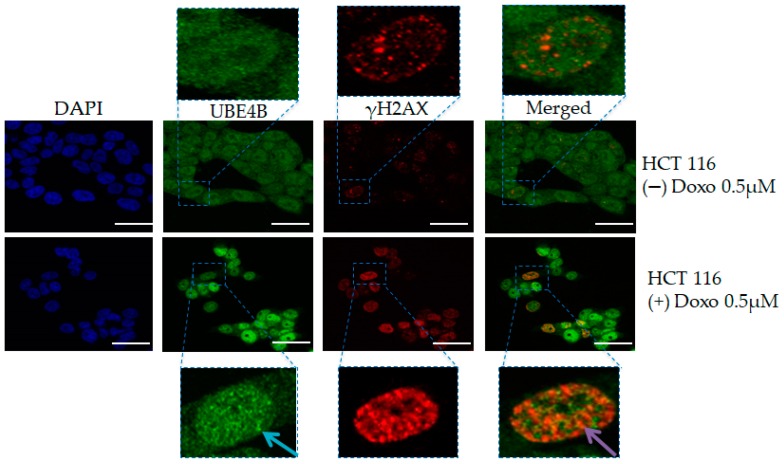
Foci UBE4B formation following DNA damage. UBE4B foci accumulate in the nucleus (blue arrow) compared to untreated cells (diffuse pattern) in HCT116 cells treated with 0.5 μΜ Doxorubicin. The staining is more intense and an extensive co-localization (orange granules) among UBE4B and γH2AX (purple arrow) is distinguished compared to untreated cells. Scale bar: 50 μM.

**Figure 2 cancers-12-00062-f002:**
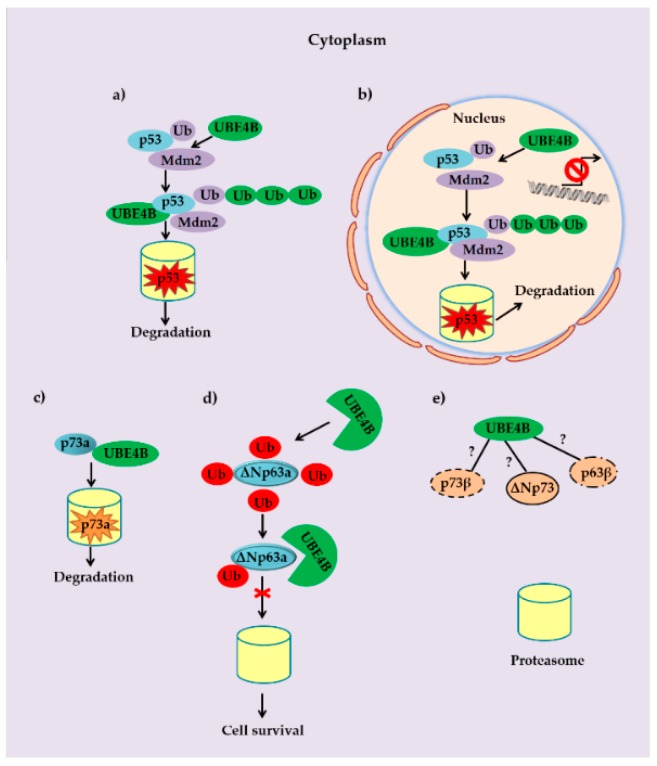
Interactions between UBE4B and p53 family members. (**a,b**) Turnover of p53 is regulated by both the MDM2 and the U-box type E3/E4 ubiquitin ligase UBE4B. Mono-ubiquitination or mono-ubiquitination of p53 at multiple MDM2 sites is usually followed by a poly-ubiquitination of these sites mediated by UBE4B. The elongated ubiquitin chain targets the modified p53 molecule to the proteasome. This sequel occurs both in the cytoplasm and the nucleus, reducing not only the protein levels of p53 but also its transcriptional activity. (**c**) Ectopically expressed UBE4B ligase in various cancer cell lines interacts physically with p73a isoform and hinders the tumor-suppressor activity of the latter by promoting its proteasomal degradation in an ubiquitination-independent manner. (**d**) Evidence regarding UBE4B’s tumorigenic role has been reported in head and neck along with lung cancer cell lines. It appears that UBE4B protects ΔΝp63a from ubiquitination, resulting in its stabilization. (**e**) The potential role of UBE4B on regulating other p53 family isoforms such as p73β, ΔΝp73 and p63β is unknown and merits elucidation.

**Figure 3 cancers-12-00062-f003:**
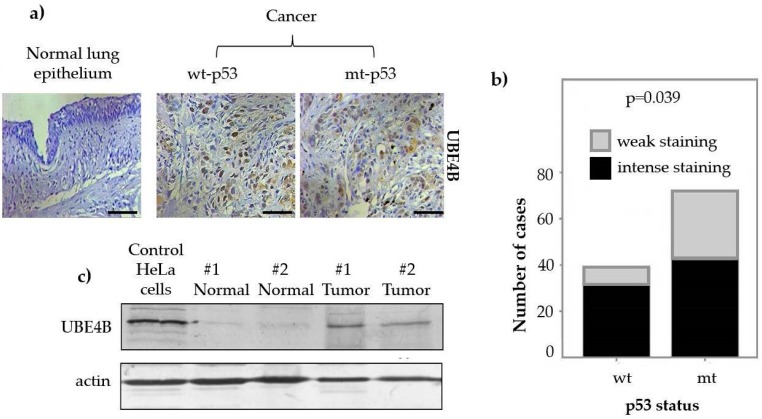
Immunohistochemistry (IHC) and Western Blot (WB) analysis of UBE4B expression in lung tissue samples. (**a**) Representative images of UBE4B IHC in normal versus lung tumor tissues with either wild type or mutant p53. Regardless of p53 status, UBE4B is overexpressed in tumor samples compared with normal ones. Mutant p53 tumor samples show less-intense UBE4B staining than in the corresponding wild type p53 tumor samples. Scale bar: 50 μm. (**b**) Statistical analysis of UBE4B IHC expression in lung carcinomas with wt-p53 (38 cases) and mut-p53 (70 cases). Wt-p53 tumor cases show a higher ratio of intense versus weak UBE4B staining (0.925) compared with mut-p53 tumor cases (0.6). (**c**) Representative immunoblots of two pairs (normal versus tumor) of lung tissue samples where UBE4B is increased in tumors with respect to their normal counterparts. HeLa cells were used as positive control.

**Figure 4 cancers-12-00062-f004:**
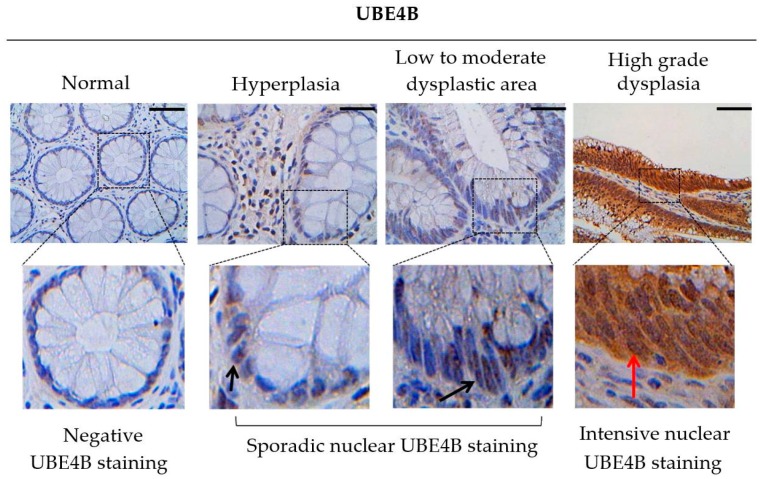
Comparison of UBE4B in colon pre-cancerous lesions in relation to normal areas. In hyperplasia and low to moderate dysplasia, UBE4B seems to be overexpressed sporadically in some nuclei of tubular glands of Lieberkühn (black arrows). In high grade dysplasia, the intensity of staining is higher (red arrow). Scale bar: 50 μm.

**Figure 5 cancers-12-00062-f005:**
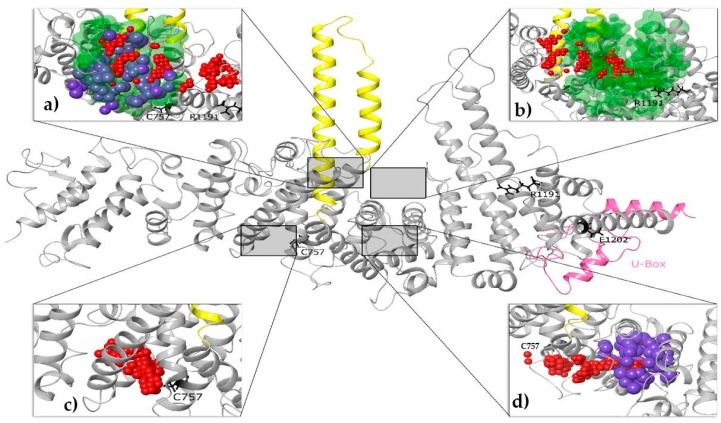
A three-dimensional structural homology model of human UBE4B, derived from comparative modelling with Ufd2p. The protein is shown in a “ribbon” representation, with a U-box domain colored pink. Important residues, found to be involved with Cdc48 binding [85,86], are depicted in a “ball and stick” representation, colored black. Selected a-helixes are colored yellow for viewing reference. Druggability analysis (likelihood of ligands binds tightly) [79,80,81] results are also illustrated in (**a**–**d**): Sitemap as small red spheres, fpocket as large blue spheres and CavityPlus as green surfaces. Only high-ranked cavities with particularly promising druggabilities were collected and compared from each program. All three algorithms converged with respect to the protein cavity shown in (**a**), whereas two out of the three suggested a promising druggability of cavities shown in (**b**,**d**). The highest druggability score (SiteScore: 1.293; D-score: 1.368; threshold for druggable sites: > 0.80) with Sitemap was observed for cavity shown in (**c**), which is adjacent to C757 residue.

**Table 1 cancers-12-00062-t001:** UBE4B in human pathology.

Data from Cellular, Animal Models and Tissues						
Pathological Condition	Cause	Role of UBE4B	Possible Alteration (s) of UBE4B	Type of Organism	Type of Tissue	Reference
Machado-Joseph disease/Spinocerebellar ataxia type 3 (SCA3)	increased CAG repeat tract in *ATXN3*	Ectopically overexpressed UBE4B poly-ubiquitinates ataxin-3 resulting in the degradation of the latter	−	Human cancer cell lines and SCA3 *Drosophila* model	−	[60]
**Scleroderma_Autoimmune disease**	−	Autoantigen in 10% of patients, regulation of mitotic progression	Phosphorylated during mitosis as a result its conformation changes dramatically and the protein doesn’t work properly leading to mitotic abnormalities	Human cancer cell lines	−	[79]
**Inclusion body myopathy with Paget disease of bone and frontotemporal dementia (IBMPFD)**	Mutant *VCP*	Interaction of mutated p97 with UBE4B weakens while its interaction with ataxin-3 becomes intensified	−	Human cancer cell lines	Muscular	[80]
**Neuroblastoma (NBL)**	*MYCN* amplification and chromosome 1p deletions	−	Deletion of the *UBE4B* gene/one base substitution in exon 9 observed in stage 3 tumor (dysfunctional protein)/low expression protein in high stage tumors with poor prognosis	NBL cell line	NBL frozen tissues, peripheral blood	[10,81]
**Oral Squamous Cell Carcinoma (OSCC)**	Multiple genetic events involving inactivation of tumor suppressor genes	−	Deletion of the *UBE4B* gene	−	Tissues from OSCC patients	[82]
**Ependymoma, Medulloblastoma**	Multiple genetic events	Negative regulation of p53 by UBE4B	Protein is frequently upregulated due to the mRNA overexpression, attributed to gene amplification	Human brain cancer cell lines and mouse model	Human tissues derived from brain tumors	[46]
**Breast Cancer**	Multiple genetic events	Negative regulation of p53 by UBE4B	Protein overexpression	Human breast cancer cell lines and mouse model	Human tissues derived from breast cancer patients	[38]
**Hepatocellular Carcinoma (HCC)**	Multiple genetic events	Negative regulation of p53 by UBE4B	Overexpression of protein and mRNA levels	Human HCC cell lines	Human tissues derived from HCC patients	[56]

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
