# Peer review of "The Role of E3, E4 Ubiquitin Ligase (UBE4B) in Human Pathologies"

_cancers, 2019, doi:10.3390/cancers12010062_

Round 1

Reviewer 1 Report

This is a great review pointing out the role of UBE4B on human cancer. The paper is very well organized, starting from the definition of ubiquitination and ubiquitin system and then going more in deep and focus on UBE4B. Particularly I appreciate the section devoted to UBE4B in human pathology because it could be a good starting point for research exploring this line. However, I suggest the authors to expand this section. I know the journal is a Cancer journal, but it will be very useful if apart from neurodegenerative diseases and cancer, the authors include other examples of the involvement of this protein in diseases, for future development of research lines. 

Thank you very much. 

Author Response

We would like to thank the reviewer for the supportive comments. Taking into consideration the reviewer’s suggestion we incorporated a table presenting the involvement of UBE4B not only in cancer but also in other disorders. The table is placed after section 4.1, and is highlighted in green. Moreover, we expanded section 4.1, by adding more information about the role of UBE4B in neurodegenerative diseases.

Reviewer 2 Report

This review by Lagopati, Antoniou et al. offers an overview of the role of the E3/E4 ubiquitin ligase UBE4B. The review focuses on the implication of UBE4B in the development of cancer, in relation to its effect on the modulation of p53 proteins. The review provides a nice summary of the current knowledge on UBE4B, and is supported by original data from the authors. Therefore, I believe that this review will be valuable to the scientific community with an interest in E4 enzymes. Hereafter are a few points that need to be clarified before publication.

Main comments:

UBE4B protein has a U-box domain that is important for its function. It would be valuable if the authors could give more details on the structure/function of this type of domain in ubiquitin ligases. MDM2 in the text but appears as HDM2 in the figure 1a. Section 4.1: the authors should comment on whether UBE4B has a beneficial or deleterious effect on the nervous system. Section 4.2: it would be beneficial if the authors mention the different mutations on UBE4B associated with cancer. Maybe a table could summarise the finding in relation to UBE4B expression/mutation and their association with cancers.

Minor comments:

Line 51: “oxidatively modified proteins” -> “oxidised proteins”?

Line 52: G76 refers to the last residue of ubiquitin molecule, which could be misleading. Keep the G76 notation when referring specifically to the ubiquitination process and referring to this particular residue.

Line 53: “b-sheet” -> the special character for “beta” was replaced by a regular “b”

Line 59: mentioning the MW of ubiquitin is not relevant here, but would make more sense when introducing the ubiquitin molecule line 51.

Lines 68-69: cite the different classes of E3s.

Line 85-86: precise that the “new class” of ubiquitination enzymes the authors referring to is the U-box domain-containing ubiquitin ligase?

Line 94: “C. elegans” should be italicised.

Line 127: “…, Alzheimer’s disease” -> “… diseases”.

Lines 144-145: the sentence starting with “Ubiquitination is a common pathway for p63…”, may be re-phrased. Is UBE4B specifically regulating the ubiquitination on p63 that promote p63’s proteasomal degradation?

Line 169: define “IR”.

Lines 179-180: the sentence starting with “A potential interaction between…” needs to be re-phrased.

Line 214: define “NSCLC”.

Line 218: “motif” -> do the authors mean “expression pattern”?

Lines 217-219: The sentence starting with “Screening for UBE4B…” is confusing and may need to be re-phrased.

Lines 245-246: sentence starting with “it has been shown that upon gamma irradiation…” -> could be re-written as “it has been shown that gamma irradiation, a known DSB inducer, leads to increased UBE4B protein levels in various human cancer cell lines”.

Line 251: “Mdm2 or Hdm2” -> “Mdm2/Hdm2”? Although, previous references to Mdm2 in this manuscript only mention it as Mdm2 and not as Hdm2; for more consistency, stick to one nomenclature.

Lines 254-253: “Hdm2” and “MDM2/Hdm2” -> use the same nomenclature/name for a specific gene/protein to avoid confusion.

Line 271: “Ube4b” -> are the authors referring to the gene or the protein name? Confusing as it appears as “UBE4B” in the rest of the manuscript.

Line 271: “Saccharomyces cerevisiae” should be italicised.

Line 320: it is unclear what “this behavior” refers to.

Author Response

Comments: This review by Lagopati, Antoniou et al. offers an overview of the role of the E3/E4 ubiquitin ligase UBE4B. The review focuses on the implication of UBE4B in the development of cancer, in relation to its effect on the modulation of p53 proteins. The review provides a nice summary of the current knowledge on UBE4B, and is supported by original data from the authors. Therefore, I believe that this review will be valuable to the scientific community with an interest in E4 enzymes. Hereafter are a few points that need to be clarified before publication.

Response: We thank the reviewer for the supportive overall evaluation.

Main comments:

UBE4B protein has a U-box domain that is important for its function. It would be valuable if the authors could give more details on the structure/function of this type of domain in ubiquitin ligases. MDM2 in the text but appears as HDM2 in the figure 1a. Section 4.1: the authors should comment on whether UBE4B has a beneficial or deleterious effect on the nervous system. Section 4.2: it would be beneficial if the authors mention the different mutations on UBE4B associated with cancer. Maybe a table could summarise the finding in relation to UBE4B expression/mutation and their association with cancers.

 Response: We thank the reviewer for this important suggestions. Relative to the first comment about the structure/function of the U-box catalytic domain, please see the additional information incorporated in the revised manuscript at lines 90-94.

Following your comment on HDM2, we adopted the term MDM2 through the whole manuscript for uniformity reasons.

Section 4.1 was further expanded to incorporate additional information about the role of UBE4B in neurodegenerative diseases. Please, see lines 237-248.

Regarding section 4.2., we expanded the text by adding more information (see lines 255-264) and incorporated a new table (before section 4.2) that summarizes the role of UBE4B in human pathology, while all known mutations affecting this ligase in cancer are included also.

Minor comments:

Line 51: “oxidatively modified proteins” -> “oxidised proteins”?

Response: Τhe correction has been included and is highlighted in green. Please see line 52.

 Line 52: G76 refers to the last residue of ubiquitin molecule, which could be misleading. Keep the G76 notation when referring specifically to the ubiquitination process and referring to this particular residue.

Response: Τhe proposed correction has been included and is highlighted in green. Please see line 53.

 Line 53: “b-sheet” -> the special character for “beta” was replaced by a regular “b”

Response: Τhe suggested correction has been included and is highlighted in green. Please see line 54.

Line 59: mentioning the MW of ubiquitin is not relevant here, but would make more sense when introducing the ubiquitin molecule line 51.

Response: The suggestion has been taken into account and the correction is highlighted in green. Please see line 53.

Lines 68-69: cite the different classes of E3s.

Response: Requested modification has been included. See lines 72-73 marked in green.

Line 85-86: precise that the “new class” of ubiquitination enzymes the authors referring to is the U-box domain-containing ubiquitin ligase?

Response: Τhe suggested correction has been included and is highlighted in green. Please see lines 95-96.

Line 94: “C. elegans” should be italicised.

Response: Τhe proposed correction has been included and is highlighted in green. Please see line 107.

Line 127: “…, Alzheimer’s disease” -> “… diseases”.

Response: Correction has been incorporated and is highlighted in green. Please see line 150.

Lines 144-145: the sentence starting with “Ubiquitination is a common pathway for p63…”, may be re-phrased. Is UBE4B specifically regulating the ubiquitination on p63 that promote p63’s proteasomal degradation?

Response: Τhe proposed correction has been included and is highlighted in green. Please see lines 167-168.

Line 169: define “IR”.

Response: IR is defined in lines 107-108. Τhe correction is highlighted in green.

Lines 179-180: the sentence starting with “A potential interaction between…” needs to be re-phrased.

Response: Τhe proposed correction has been included and is highlighted in green. Please see the lines 203-205.

 Line 214: define “NSCLC”.

Response: Definition of abbreviation has been added. Please see line 271.

Line 218: “motif” -> do the authors mean “expression pattern”?

Response: Τhe proposed correction has been included and is highlighted in green.Please see line 275.

Lines 217-219: The sentence starting with “Screening for UBE4B…” is confusing and may need to be re-phrased.

Response: Τhe sentence has been corrected and is highlighted in green Please see lines 274-277.

Lines 245-246: sentence starting with “it has been shown that upon gamma irradiation…” -> could be re-written as “it has been shown that gamma irradiation, a known DSB inducer, leads to increased UBE4B protein levels in various human cancer cell lines”.

Response: Τhe sentence has been re-written and correction is highlighted in green Please see lines 292-293.

Line 251: “Mdm2 or Hdm2” -> “Mdm2/Hdm2”? Although, previous references to Mdm2 in this manuscript only mention it as Mdm2 and not as Hdm2; for more consistency, stick to one nomenclature.

Response: Thank you for bringing to our attention this point. We adopted MDM2 as a uniform nomenclature throughout the text.

Lines 254-253: “Hdm2” and “MDM2/Hdm2” -> use the same nomenclature/name for a specific gene/protein to avoid confusion.

Response: See previous response.

Line 271: “Ube4b” -> are the authors referring to the gene or the protein name? Confusing as it appears as “UBE4B” in the rest of the manuscript.

Response: Please see the relative adjustments in lines 331, 333, 350, 351, 354, 356.

Line 271: “Saccharomyces cerevisiae” should be italicised.

Response: Τhe proposed correction has been incorporated and is highlighted in green. Please see line 331.

Line 320: it is unclear what “this behavior” refers to.

Response: Τhe sentence has been corrected and is highlighted in green. Please see lines 380-381.

Reviewer 3 Report

The Authors explored the role of UBE4B by reviewing its role in physiological and pathological conditions and adding some original data to the manuscript. The manuscript is well written and easy to follow although there are some errors/typos that should be corrected. Besides, through the manuscript I have some concerns outlined here below.

As the title indicates, the manuscript is mainly focused on the implication of UBE4B in cancer. I believe the mini-section on neurodegenerative disease on page 4 as well as the ataxin 3 involvement (page 3 lines 131-135) should be deleted.

I would expand instead on: a) the Ubiquitin fusion degradation pathway UFD; and b) the biochemical role of UBE4B, which are not described.

Figure 1. The sub-panels should be numbered according to their citation order in the text.

Page 4, line 148, ‘This stabilization was first noted…’ is not clear and should be better explained.

Page 4, line 157 ‘proteosomal degradation without any intermediate ubiquitination’ should be better elaborated as well.

Minor points: (G76), line 52 is not appropriate; ‘on K48 residue’ line 60 should read ‘on K48 residue of ubiquitin’ ‘The role of E3 ligases in carcinogenesis is not entirely clear’, line 69 is unnecessary; The sentence starting on page 2 line 95, ‘Thus, a ubiquitin…’ is not clear and should be rephrased.

Author Response

Comments: The Authors explored the role of UBE4B by reviewing its role in physiological and pathological conditions and adding some original data to the manuscript. The manuscript is well written and easy to follow although there are some errors/typos that should be corrected. Besides, through the manuscript I have some concerns outlined here below.

Response: We would like to thank the reviewer for the supportive comments. Minor typos have been corrected.

As the title indicates, the manuscript is mainly focused on the implication of UBE4B in cancer. I believe the mini-section on neurodegenerative disease on page 4 as well as the ataxin 3 involvement (page 3 lines 131-135) should be deleted.

Response: Thank you very much for your suggestion. Since the implication of UBE4B is also important in various human pathologies, apart from cancer, we modified the title of this review paper to cover pathology. Thus, we kept the information about the neurodegenerative diseases as well as the ataxin 3 involvement, since expansion of this point was requested by the other reviewers also. Furthermore, we incorporated a table (among sections 4.1 and 4.2) which summarizes the role of UBE4B in a variety of pathological conditions, including different types of cancer.

I would expand instead on: a) the Ubiquitin fusion degradation pathway UFD; and b) the biochemical role of UBE4B, which are not described.

Response: Thank you for this important suggestion. We expanded accordingly on this subject. Please see lines 86-88, 90-94 and 101-105 in the revised manuscript. The corrections are highlighted in green.

 Figure 1. The sub-panels should be numbered according to their citation order in the text.

Response: Please see the last revised paragraph of section 3. Please also see revised figure 2 (previous figure 1).

Page 4, line 148, ‘This stabilization was first noted…’ is not clear and should be better explained.

Response: The suggested revision has been expanded and corrections are highlighted in green. Please see lines 171-172.

Page 4, line 157 ‘proteosomal degradation without any intermediate ubiquitination’ should be better elaborated as well.

Response: Please see lines 180-181. The corrections are highlighted in green.

 Minor points: (G76), line 52 is not appropriate; ‘on K48 residue’ line 60 should read ‘on K48 residue of ubiquitin’ ‘The role of E3 ligases in carcinogenesis is not entirely clear’, line 69 is unnecessary; The sentence starting on page 2 line 95, ‘Thus, a ubiquitin…’ is not clear and should be rephrased.

Response: Please see lines 53 (G76 has been omitted), 60-62, 73 (the sentence ‘The role of E3 ligases in carcinogenesis is not entirely clear’ has been deleted), 108-110. Corrections are highlighted in green.

Round 2

Reviewer 2 Report

In their updated version of the manuscript, the authors have made significant changes and taken into account each of the comments made previously.

One minor comment remains:

The authors have added a new figure (now Figure 1). In insets, the authors show a zoomed-in image of one single representative cell; however, for the condition where cells were treated with Doxo, the cell in the inset showing the UBE4B staining does not correspond to the cell shown for the gammaH2AX and merged panels. Like for the untreated cells, all the inset should show the zoomed-in image for one same cell.

Author Response

Thank you for this critical observation. We have updated figure 1 showing now the appropriate inset magnifications as suggested (see the updated figure 1).